# MKA: Leveraging Cross-Lingual Consensus for Model Abstention

**Sharad Duwal**
sharad.duwal@gmail.com

## Abstract

Reliability of LLMs is questionable even as they get better at more tasks. A wider adoption of LLMs is contingent on whether they are usably factual. And if they are not, on whether they can properly calibrate their confidence in their responses. This work focuses on utilizing the multilingual knowledge of an LLM to inform its decision to abstain or answer when prompted. We develop a multilingual pipeline to calibrate the model's confidence and let it abstain when uncertain. We run several multilingual models through the pipeline to profile them across different languages. We find that the performance of the pipeline varies by model and language, but that in general they benefit from it. This is evidenced by the accuracy improvement of 71.2% for Bengali over a baseline performance without the pipeline. Even a high-resource language like English sees a 15.5% improvement. These results hint at possible further improvements.

## 1 Introduction

While large language models have driven extensive progress in natural language processing tasks, their utility still hinges on their reliability. Reliability of LLMs is determined by their factuality and their tendency to "hallucinate" when relevant information is not available (as parametric or non-parametric knowledge) (Mishra et al., 2024; Asai et al., 2024; Mallen et al., 2023). LLMs are, by design, prone to hallucinations. And hallucinations are a major concern for LLMs deployed in the wild, especially in safety-critical situations like the medical domain (Ji et al., 2023).

Hallucination is an architectural limitation and feature of current language modeling tools. When faced with uncertainty, instead of providing misleading responses, a desirable action for an LLM is to abstain from answering. Research in LLM reliability through abstention has been focused on implementing methods at different stages of training (pretraining, finetuning, post-training) and during inference (Wen et al., 2024).

One line of hallucination abstention research is directed toward calibration of model confidence. This is achieved by utilizing the knowledge of the generating model itself (Kadavath et al., 2022; Jiang et al., 2021; Feng et al., 2024b; Xiong et al., 2024) to quantify its confidence on its answer. Using a confidence threshold, the model can then be made to abstain from responding, or to respond with caution.

In this work, we approach the abstention question using the multilingual knowledge of the model. We design an inference pipeline with the assumption that it is possible to meaningfully apply a model's knowledge in several languages to check the correctness of its answer in any one language.

Multilingual knowledge of LMs has been used before for abstaining by having the model provide feedback on its own answer (Feng et al., 2024a), similar to the self-reflection method in (Ji et al., 2023). Instead of using feedback to improve an answer (which is better suited for open-ended generation tasks), we translate questions prompt the model separately, then translate the answers back to the original language and arrive at a consensus.

Etxaniz et al. (2023) find that language models cannot utilize cross-lingual knowledge implicitly: prompting in one language doesn't necessarily utilize the model's knowledge in other languages, i.e., language models are unable to leverage multilingual knowledge if monolingually prompted. Thus, in order to utilize the cross-lingual knowledge of a model, we use more than one language (grouped

by categories like resource level, language family, etc.) to calibrate the model's confidence to make abstentions.

We introduce the MKA (Multilingual Knowledge Abstention) pipeline where given a question in a certain ("target") language, we translate it to a group of related ("auxiliary") languages to prompt a multilingual model to get responses in these languages, which are translated back to the target and then utilized to calibrate the model's confidence based on the semantic similarity of the responses.

## 2 METHOD

**Problem Formulation** Given a language model $f : P \to R$ where $P$ and $R$ are prompt and response spaces, model abstention is a function $g : (p, r) \to \{0, 1\}$ where $p \in P$, $r \in R$. The model abstains if $g = 1$. The function $g$ for this work is the MKA pipeline.

### 2.1 MKA PIPELINE

The MKA pipeline is an inference-time confidence-calibration method. We use knowledge-based MCQA (multiple-choice question answering) benchmarks for our experiments. The evaluation set has triples comprising questions, choices and answers: $(q, c, x)$. For illustration purposes, let's consider the evaluation set is in language $t$ and the auxiliary language set is the low-resource $LR$ set. Steps for the MKA pipeline are as follows:

1. **Translation:** We use a translation system to translate the question $q$ and the choices $c$ (in target language $t$) into languages as grouped in different auxiliary language sets. For example, for $LR = \{LR_0, LR_1, LR_2, \ldots\}$, we translate $q$ into all these languages. Our low resource set $LR$ includes Telugu, Nepali, Maithili, Bhojpuri, Yoruba, and Zulu.

2. **Prompting:** Once $q$ and $c$ have been translated into the auxiliary languages of choice, we construct the prompts $P_{LR} = \{p_0, p_1, p_2, p_3, \ldots\}$ by concatenating the translated questions and choices and adding prompting instructions. Then we prompt the model $f$ with all prompts in $P_{LR}$.

   Once we have the model's generations, we process them to get only the answers $R_{LR} = \{r_0, r_1, r_2, \ldots\}$. We assume all answers are in the prompting language (not necessarily $t$). To standardize the answers in order to compare them with the true answer $x$, we translate all answers back to the target language $A_{LR} = \{a_0, a_1, a_2, \ldots\}$.

3. **Cosine Similarity Centroid Polling:** We poll all answers in $A_{LR}$ to select the answer $a_s$ with the highest average cosine similarity across all answers. Using character n-grams as features to construct vectors $v_i$ and $v_j$ for answers $a_i$ and $a_j$, we consider as the model's final answer the response at position

$$i^* = \underset{i \in \{1, \ldots, n\}}{\arg\max} \left( \frac{1}{n} \sum_{j=1}^{n} \text{cos\_sim}(v_i, v_j) \right)$$

$$\text{where } \text{cos\_sim}(v_i, v_j) = \frac{v_i \cdot v_j}{\|v_i\| \|v_j\|}$$

4. **Confidence calibration:** Then we find the confidence of the model on the selected answer $a_s$ using the cosine similarity between sentence embeddings of the other answers with that of $a_s$. For answers that have a similarity greater than $0.8$, we multiply their confidence weight by $1.5$ (assumption: few very similar answers provide more confidence than many dissimilar answers). This calibration method is model-independent. Given sentences $a_s$ and $a_i$, for sentence embeddings $e_s$ and $e_i$ computed using the embedding model `paraphrase-multilingual-mpnet-base-v2` (Reimers & Gurevych, 2020), confidence is calculated as

$$c_f = \frac{1}{|A \setminus a_s|} \sum_{i=1}^{|A \setminus a_s|} w_i \, \text{cos\_sim}(e_s, e_i)$$

$$\text{where } w_i = \begin{cases} 1.5 & \text{if cos\_sim}(e_s, e_i) > 0.8 \\ 1 & \text{otherwise} \end{cases}$$

Finally, we use a confidence cutoff $c_c$ to decide whether to abstain or answer given model confidence $c_f$ on answer $a_s$.

## 2.2 BASELINE

To evaluate the performance of the MKA pipeline, we also benchmark the language models without the pipeline. For this, we prompt the language model in the target language for all questions from the evaluation set and use the same evaluation method and metrics as MKA. Refer to Table 2 for the baseline scores.

## 2.3 AUTOMATIC EVALUATION

Even though we are working with an MCQA task where the model has only to choose an option from given choices, we cannot use exact matching because a model answer is translated to the target language from some auxiliary language, which often introduces artifacts. Thus, to evaluate responses, we calculate the semantic similarity between the correct answer and the model answer as agreed on by the MKA pipeline. Specifically, we use the cosine similarity and apply a cut off of $0.85$ to establish if the model answer is correct. Once again we use the `paraphrase-multilingual-mpnet-base-v2` model offered by SentenceTransformers to get sentence embeddings (Reimers & Gurevych, 2020) for evaluation.

## 3 EXPERIMENTS

### 3.1 SETUP

**Models.** For translation, we use an `int8`-quantized version of the `NLLB-200 1.3B Distilled` model (Team et al., 2022). For prompting, we focus on models: `Aya Expanse 8B` (Dang et al., 2024), `Gemma 2 9B` (Team et al., 2024), `Qwen 2.5 7B` (Qwen et al., 2025) and `Gemma 2 2B` (Team et al., 2024). To investigate how the model size correlates to the pipeline performance, we also experiment with the `Gemma 2 27B` (Team et al., 2024) to enable a comparison across the three models from the Gemma 2 family (Appendix A). We use SGLang (Zheng et al., 2024) to prompt the models.

**Eval sets.** Because we are focusing on MCQA, we use the multilingual MMLU (OpenAI, 2024; Hendrycks et al., 2021). Among the languages available we evaluate the MKA pipeline on prompts originating in six target languages: Bengali, English, Swahili, Yoruba, Japanese and Indonesian.

**Auxiliary language sets.** We establish three auxiliary language sets based on resource levels: *high-resource* (English, German, French, Spanish, Simplified Chinese, Portuguese), *mid-resource* (Greek, Hebrew, Hindi, Indonesian, Ukrainian, Vietnamese) and *low-resource* (Telugu, Nepali, Maithili, Bhojpuri, Yoruba, Zulu).

All questions and answers will be in the target languages while the intermediate processing will be done in the auxiliary languages.

To evaluate the performance of the MKA pipeline over the baseline method (§2.2), we calculate the pipeline's confidence on the answers agreed upon by the model responses using the centroid polling method (§2.1.3). Then we use confidence categories to establish whether there are definitive confidence cutoffs (from 0 to 1 in increments of $0.02$, i.e., $50$ categories) where the pipeline is better for specific configuration of target language, auxiliary languages and prompting model. To establish these confidence categories, we find the average performance of a prompting model across its MKA runs (total target language $\times$ auxiliary language sets runs). We consider the confidence category that has the highest average accuracy to be the optimal confidence cutoff for a model.

### 3.2 METRICS

**Accuracy.** Given our focus on MCQA-type questions, we can expect any of four different outcomes for the MKA pipeline on a question: abstain when model answer is correct (**A1**), abstain when model

answer is incorrect (**A2**), answer when model answer is correct (**A3**) and answer when model answer is incorrect (**A4**).

For this configuration we have two accuracies: abstained accuracy $AC_{abs}$ and answered accuracy $AC_{ans}$.

$$AC_{abs} = \frac{A2}{A1 + A2}$$

$$AC_{ans} = \frac{A3}{A3 + A4}$$

These metrics consider abstentions and responses individually. These accuracies therefore do not allow meaningful comparison: the ratio of abstentions-responses changes depending on the confidence cutoff, which causes the denominators to change, making the accuracies not directly comparable.

We thus use a composite metric $AC_{comp}$ that looks at both abstentions and answers at the same time and also has a constant denominator, allowing meaningful comparisons:

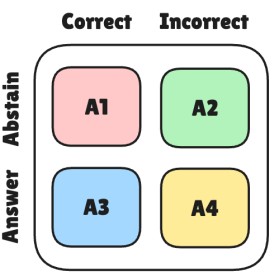

Figure 1: MKA Pipeline confusion matrix

$$AC_{comp} = \frac{\text{correctly answered + correctly abstained}}{\text{total}}$$
$$= \frac{A2 + A3}{A1 + A2 + A3 + A4}$$

However, a limitation of this composite metric is immediately obvious. A badly performing model that hypothetically answers only one question (correctly) and abstains all other questions will have an $AC_{comp}$ of 1. Thus, we also use the answer rate of the model to factor in the coverage across the eval set. We use the answer coverage of the prompting model to devise an effective accuracy:

$$AC_{eff} = AC_{comp} * coverage$$
$$\text{where } coverage = \frac{A3 + A4}{\text{total}}$$

Even this metric has the inherent bias that answering is preferable to abstaining. One way to curtail the coverage bias on the effective accuracy would be to add an exponential weight. But since we introduced the coverage part only to handle poorly performing models, we use the weight-less formulation of the effective accuracy. All accuracy reported here, unless noted otherwise, are effective accuracy.

**Accuracy v/s Coverage.** Even after introducing a multiplicative coverage component to the effective accuracy, to make the impact of the coverage more explicit, we visualize how the answered accuracy of the models is related to the answer rate (coverage) for every auxiliary language set.

## 4 RESULTS

**Model Performance and Optimal Confidence Cutoffs.** From Table 1, we can see that different models achieve optimal performance at different confidence cutoffs. To establish optimal cutoffs ($c_c$), we calculate the mean accuracy for every model across all its MKA runs. The cutoff with the highest average accuracy is used as the $c_c$ for the model.

Aya Expanse 8B and Gemma2 27B (int4) perform better than the other models overall. No model is definitively better than the others on any of the three auxiliary sets: the Gemma2 27B is usually better at the mid- and high-resource auxiliary sets, and Gemma2 9B and Aya Expanse 8B score the best accuracy on four auxiliary-language configurations each.

| Tgt. | $c_c$ | Expanse 8B
0.7 | Qwen2.5 7B
0.58 | Gemma2 2B
0.66 | Gemma2 9B
0.64 | G2 27B (int4)
0.64 |
|---|---|---|---|---|---|---|
| Ben | Low | 0.3443 | 0.3101 | 0.3117 | 0.3649 | **0.4195** (+71.2%) |
|  | Mid | 0.3280 | 0.3753 | 0.3226 | **0.4141** | 0.4025 |
|  | High | 0.3237 | 0.3675 | 0.3268 | 0.3864 | **0.3966** |
|  | *Avg* | 0.3320 | 0.3510 | 0.3204 | 0.3885 | **0.4062** |
| Eng | Low | 0.3507 | 0.3103 | 0.3450 | 0.4356 | **0.4788** |
|  | Mid | 0.4730 | 0.3245 | 0.3885 | 0.5154 | **0.5558** |
|  | High | 0.5399 | 0.5135 | 0.4307 | 0.6452 | **0.7162** (+15.5%) |
|  | *Avg* | 0.4545 | 0.3828 | 0.3881 | 0.5321 | **0.5836** |
| Yor | Low | **0.2931** | 0.2366 | 0.2828 | 0.2599 | 0.2758 |
|  | Mid | 0.2581 | 0.2451 | 0.2534 | 0.2654 | **0.2665** |
|  | High | **0.2941** (−24.6%) | 0.2403 | 0.2508 | 0.2465 | 0.2363 |
|  | *Avg* | **0.2818** | 0.2407 | 0.2623 | 0.2573 | 0.2595 |
| Swa | Low | 0.2736 | 0.2925 | 0.2593 | **0.3157** | 0.2760 |
|  | Mid | **0.3424** (+20.1%) | 0.3022 | 0.2295 | 0.3349 | 0.2958 |
|  | High | **0.3321** | 0.2950 | 0.2616 | 0.2806 | 0.3015 |
|  | *Avg* | **0.3160** | 0.2966 | 0.2501 | 0.3104 | 0.2911 |
| Jpn | Low | 0.2904 | 0.2957 | 0.3136 | **0.4070** | 0.3700 |
|  | Mid | 0.3575 | 0.3434 | 0.2943 | 0.4004 | **0.4366** |
|  | High | 0.3570 | 0.3996 | 0.3422 | 0.4246 | **0.4551** (−11.6%) |
|  | *Avg* | 0.3350 | 0.3462 | 0.3167 | 0.4107 | **0.4206** |
| Ind | Low | 0.3304 | 0.3005 | 0.3023 | **0.4433** | 0.4290 |
|  | Mid | 0.4055 | 0.3503 | 0.3173 | 0.4750 | **0.5176** |
|  | High | 0.4175 | 0.4189 | 0.3635 | 0.4931 | **0.5281** (+21.4%) |
|  | *Avg* | 0.3845 | 0.3566 | 0.3277 | 0.4705 | **0.4916** |

Table 1: **Effective accuracies of the models with the MKA pipeline using the best confidence cutoff** $c_c$ (refer to Appendix C). {Low, Mid, High} are the auxiliary language sets. **Bold** is the highest accuracy across the row. The highest accuracy for every target language also states (inside parenthesis) the percent change over the the baseline method's best accuracy for that language (Table 2). Performed on an evaluation set of size $n = 200$. **G2** is Gemma2.

| | Aya Expanse
8B | Qwen2.5
7B | Gemma2
2B | Gemma2
9B | Gemma2 27B
(int4) |
|---|---|---|---|---|---|
| Ben | 0.16 | **0.245** | 0.055 | 0.105 | 0.155 |
| Eng | 0.51 | 0.435 | 0.405 | 0.51 | **0.62** |
| Yor | **0.39** | 0.325 | 0.08 | 0.12 | 0.335 |
| Swa | **0.285** | 0.205 | 0.11 | 0.14 | 0.245 |
| Jpn | **0.515** | 0.41 | 0.245 | 0.285 | 0.45 |
| Ind | **0.435** | 0.365 | 0.215 | 0.345 | **0.435** |

Table 2: Accuracy of the models without the MKA pipeline (baseline method) on the target languages. Sample size $n = 200$.

Low-resource target languages seem to benefit from our method: the best performance in Bengali using the pipeline is 71.2% better than the best performance with the baseline method (refer to Table 2). Yoruba, another low-resource language, however, has 24.6% lower accuracy with the MKA method. This might be explained by the poor translation performance for Yoruba: NLLB reports a spBLEU of 26.6/13.8 (Yor→Eng/Eng→Yor). Compare these scores with Swahili's: 48.1/37.9.

Bengali, English, Swahili and Indonesian all benefit from the MKA pipeline. Yoruba and Japanese do not.

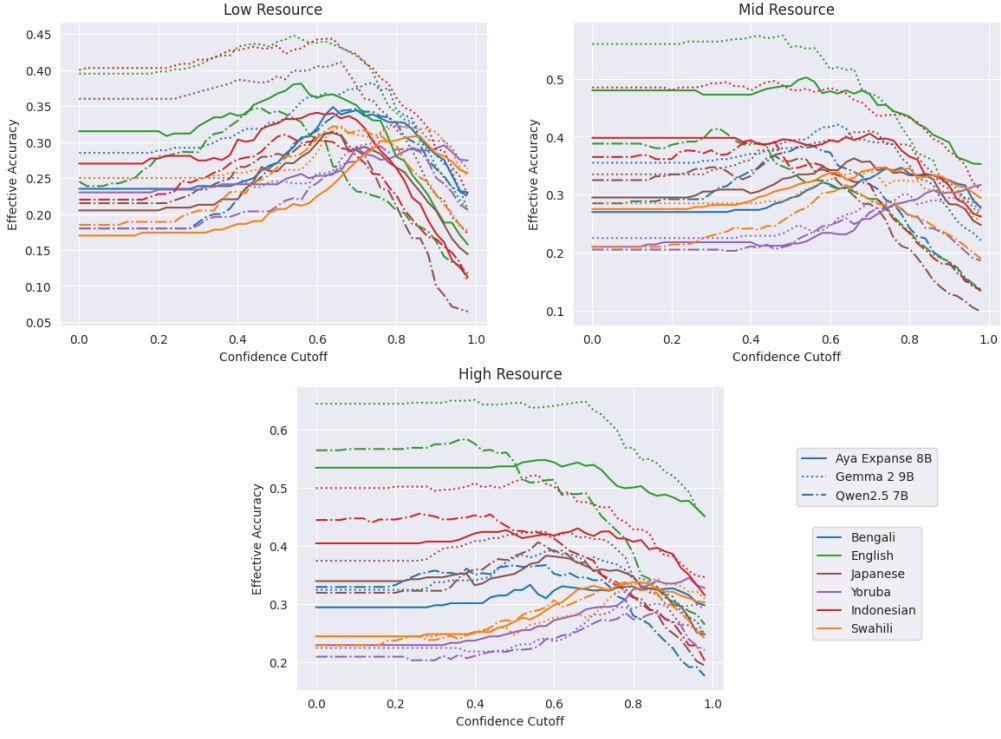

(a) Effective Accuracy versus Confidence Cutoff

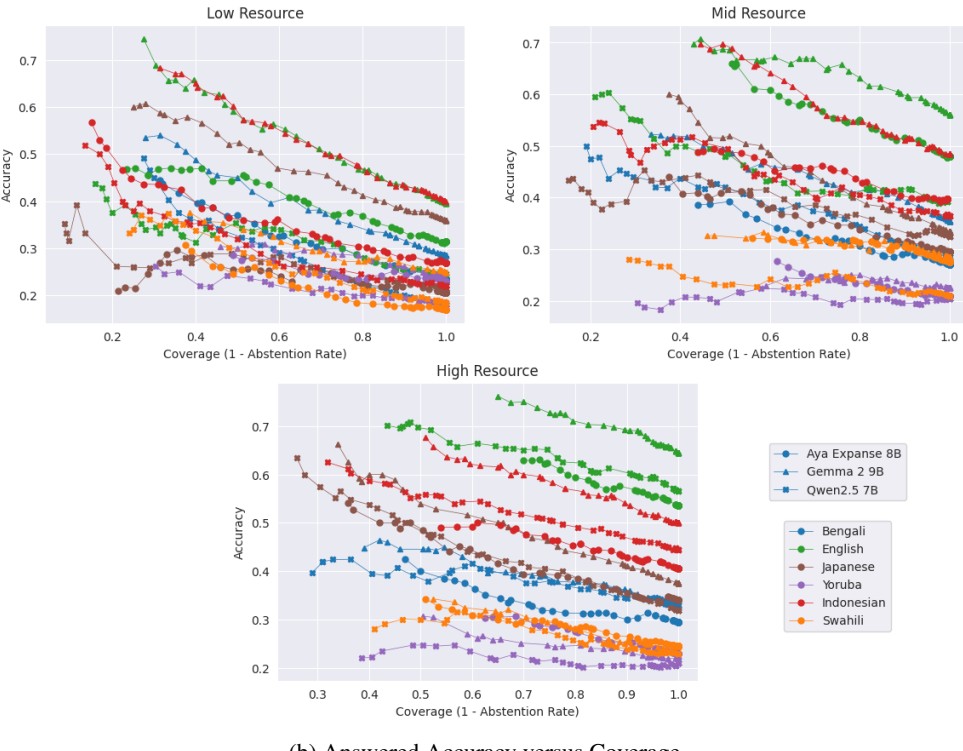

(b) Answered Accuracy versus Coverage

Figure 2: **Accuracy analyses for the MKA Pipeline** on Aya Expanse 8B, Gemma2 9B and Qwen2.5 7B across the target languages.

These results show that the MKA pipeline does not seem to depend on the target languages. Instead, it depends on the auxiliary languages and how well the translation system performs for a particular translation pair.

**Coverage and Accuracy Trade-offs.** The accuracy-coverage relationship (Figure 2b) shows the correlation between the answered accuracy and the answer rate (coverage). Higher coverage rates (lesser abstention), obviously, lead to lesser accuracy. This is most pronounced for the low-resource auxiliary languages, directly related to the reduced quality of prompts due to worse translation for these languages.

Going from the low-resource to high-resource tasks, we see the languages form gradations so that the highest-resourced languages perform well for all prompting models and the low-resource languages perform below par, but still remotely better than the baseline. We can focus on the accuracy of each model at full coverage to get a sense of this improvement. (Yoruba and Swahili start below $0.2$ in the low-resource set while they start near $0.2$ for the high-resource set).

**Prompting instruction volatility** The pipeline was observed to be volatile to the prompting instructions, especially for the `Qwen-2.5 7B` model. This is likely related to the models themselves.

## 5   CONCLUSION

We implemented an inference-time confidence calibration pipeline for LLMs that utilizes the multilingual knowledge of the model to decide whether it should answer or abstain when prompted. We evaluated the pipeline using accuracy metrics based on answers and abstentions. We benchmarked the models using a baseline method and compared its performance against the proposed method and found that given availability of good machine translation systems, the MKA pipeline can improve the accuracy of an LLM by abstaining when it lacks confidence in the responses instead of confabulating. This validates confidence calibration with multilingual knowledge as a useful tool to tackle model hallucination.

## 6   LIMITATIONS AND FUTURE WORK

The pipeline relies on semantic distance in the form of cosine similarity between sentence embeddings to evaluate the correctness of the model answers. This may be improved by using commercial LLMs as evaluator of equivalence of the model answers and the ground truths.

An interesting direction for future work would be to use the multilingual knowledge internally, i.e., without translating between the target and auxiliary languages. Such work will address many weaknesses of the current pipeline, like translation artifacts, dependence on cosine similarity, and the overall reliance on disjointed post-inference techniques.

**Reproducibility Statement** We have tried to ensure that the results in this work are reproducible by making the experiments deterministic using low temperature while decoding and using random seed when possible. The code will be available at `github.com/sharad461/MKA-hallucination`. The random seed used for the reported experiments is 97 and the sample size is 200. As discussed earlier (in Results), prompting instructions may cause variation. For the same prompt we have observed consistent results across multiple runs.

## ACKNOWLEDGMENTS

This work is part of the meta-study for the AI Researcher Project (Si et al., 2024) at Stanford NLP Group and was supported by them. We also thank Zhaofeng Wu, who originally contributed the idea for this work.

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

# A  GEMMA MODELS

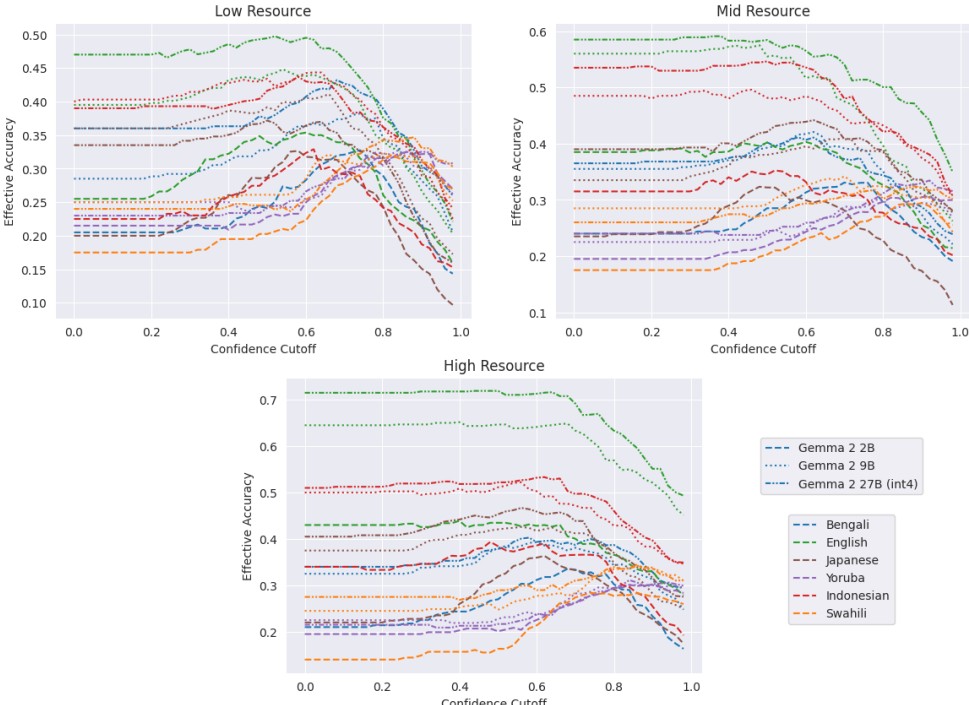

(a) **Effective Accuracy versus Confidence Cutoff** for the MKA pipeline on the Gemma models.

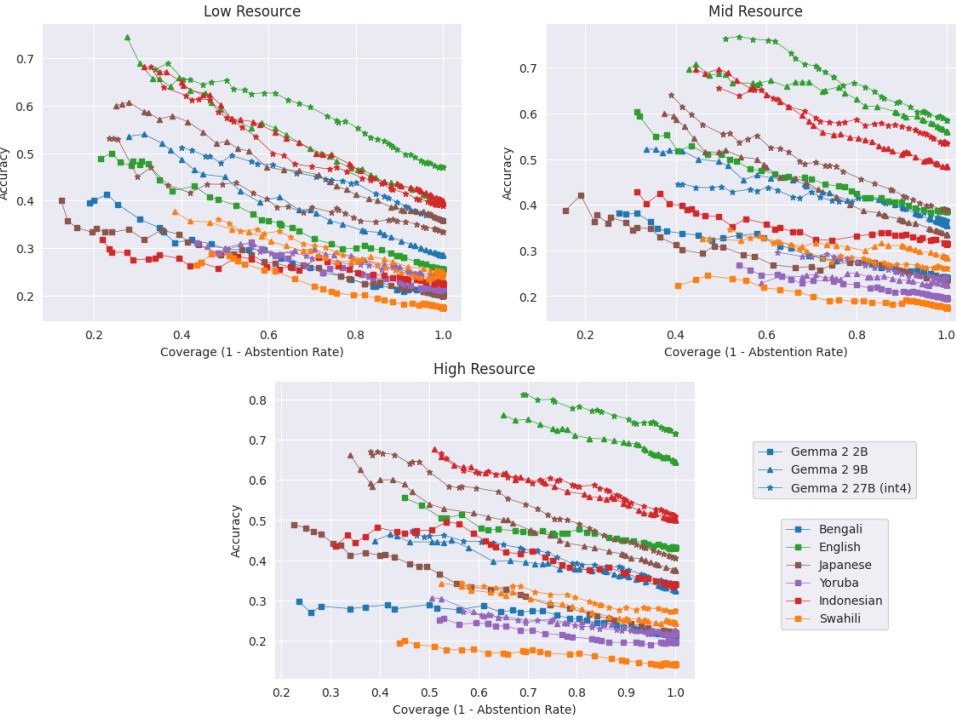

(b) **Answered Accuracy versus Confidence Cutoff.**

# B  MKA Pipeline Output Example

| Stage | Content | Language | Similarity |
|---|---|---|---|
| Original Question | At birth, the least developed part of the brain is the 
 *Options*: visual system, cortex, brain stem, cerebellum 
 (*Answer*: cortex) | English | |
| Translated Prompts | Bei der Geburt ist der am wenigsten entwickelte Teil des Gehirns die 
 *Options*: Synapse, Kortex, Hirnstamm, Zerebellum | German | |
| | À la naissance, la partie du cerveau le moins développée est le 
 *Options*: système visuel, cortex, tronc cérébral, cervelet | French | |
| | Al nacer, la parte menos desarrollada del cerebro es el 
 *Options*: sistema visual, corteza, tronco cerebral, cerebelo | Spanish | |
| | 在出生时,大脑最不发达的部分是 
 *Options*: 视觉系统, 皮层, 脑干, 小脑 | Chinese | |
| | No nascimento, a parte menos desenvolvida do cérebro é a 
 *Options*: sistema visual, córtex, tronco cerebral, cerebelo | Portuguese | |
| Model Responses | cortex | English | |
| | Kortex | German | |
| | [cortex] | French | |
| | [corteza] | Spanish | |
| | 皮层 | Chinese | |
| | córtex | Portuguese | |
| Translated Responses | **the cortex** | English | - |
| | The cortex | German | 1.000 |
| | [Cortex] What is it? | French | 0.496 |
| | [crust] | Spanish | 0.185 |
| | The cortex | Chinese | 1.000 |
| | the cortex | Portuguese | 1.000 |
| Final Decision | "the cortex" (Confidence: 1.000, Similarity with ground truth: 0.956) | | **Correct** |

Table 3: **The MKA pipeline processing a multiple-choice question.** The target language is English and the auxiliary language set is the high-resource set (German, English, French, Spanish, Simplified Chinese and Portuguese). **Bold** is the answer selected by the centroid polling method (2.1.3). The system shows high confidence and agreement across the languages. The final decision achieves high similarity with the ground truth. Thus, the model chooses to answer.

# C CONFIDENCE CUTOFF ANALYSES

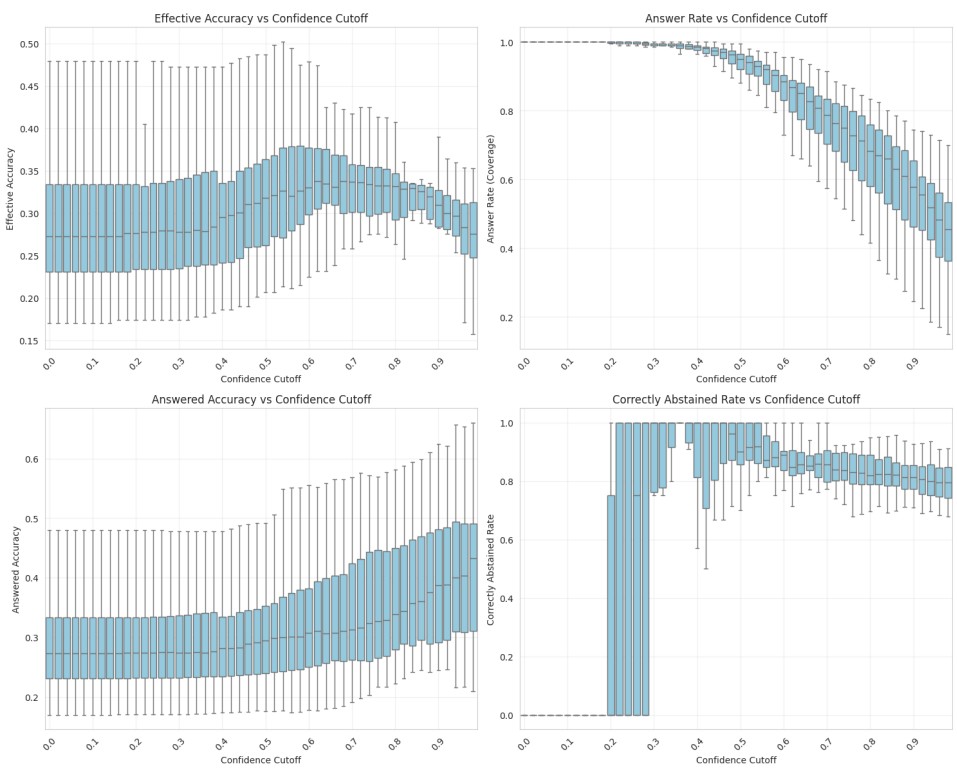

Confidence Cutoff Analysis for **Aya Expanse 8B**

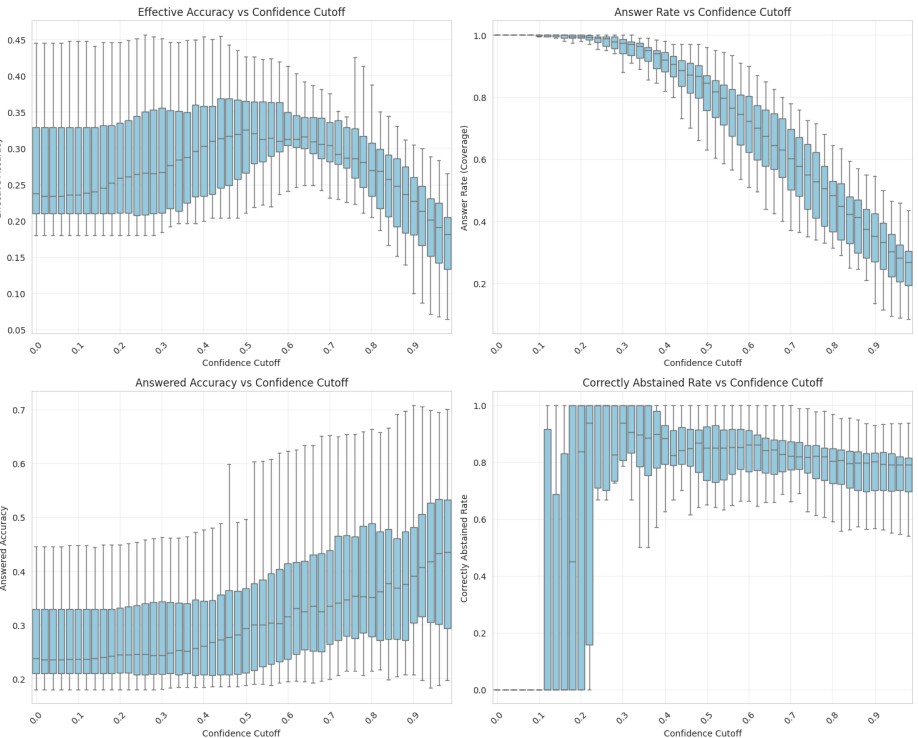

Confidence Cutoff Analysis for **Qwen2.5 7B**

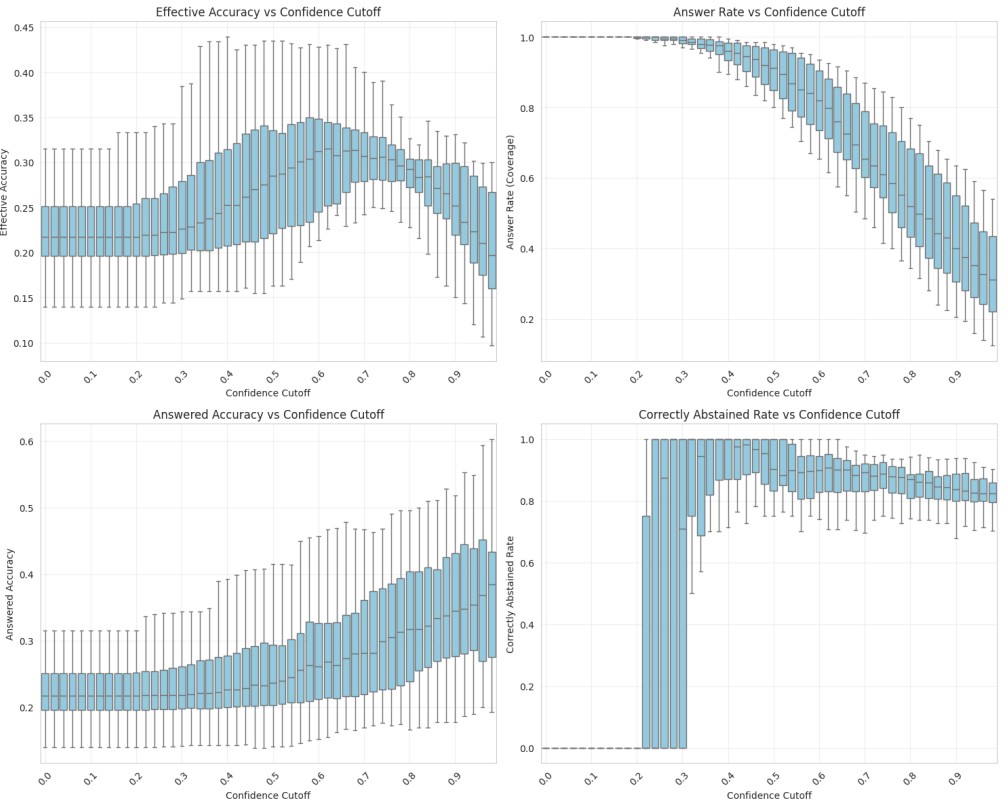

Confidence Cutoff Analysis for **Gemma2 2B**

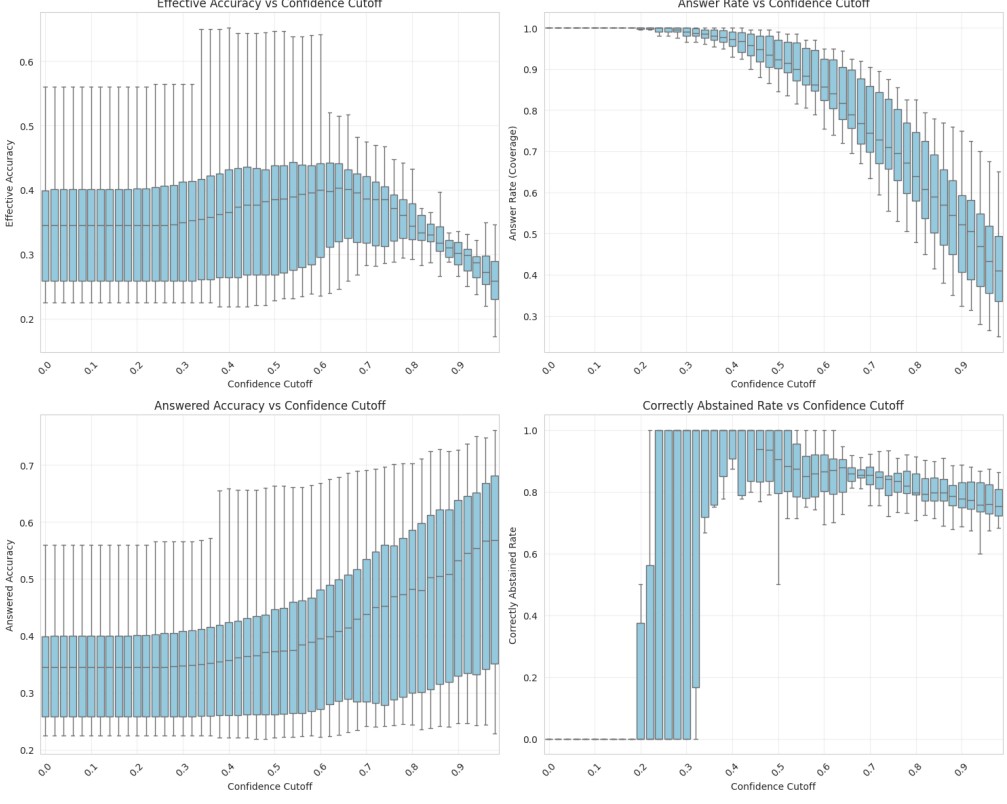

Confidence Cutoff Analysis for **Gemma2 9B**

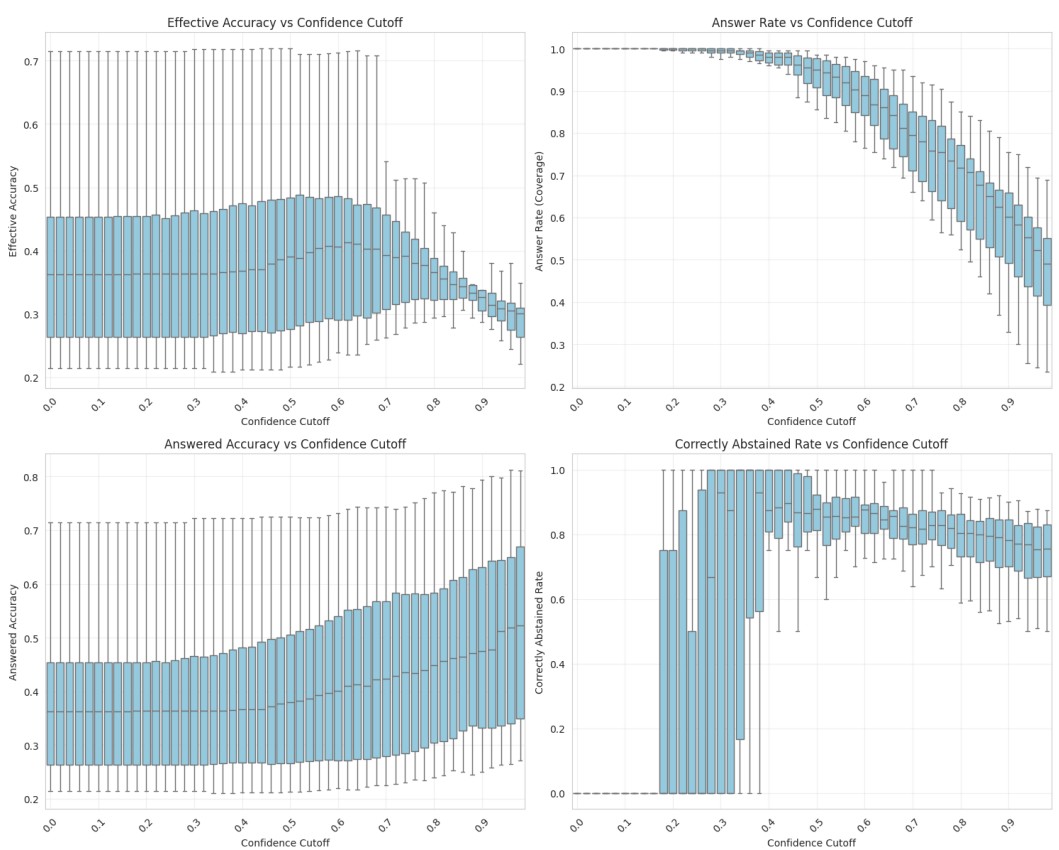

Confidence Cutoff Analysis for **Gemma2 27B**

