# OpenReview forum: "MKA: Leveraging Cross-Lingual Consensus for Model Abstention"
_ICLR.cc/2025/Workshop/BuildingTrust — BuildingTrust_

### Official Review · Reviewer_yfAc · 2025-02-16

**Rating:** 4
**Confidence:** 4

**Review:**

The paper focuses on the problem of getting LLMs to properly qualify their responses based on their confidences, and abstaining entirely when they are not confident of their response. This is a valuable area of research, and very relevant to the topic of the workshop. The paper builds on the existing work of Feng et al (2024a) and Feng et al (2024b), which suggest estimating confidence via “cross-lingual consensus”, i.e. prompting the LLM in multiple languages and seeing if they agree.

However, the contribution of this paper over these previous works does not seem significant to me. The main contribution claimed by the paper is the “MKA pipeline”, a systematic implementation of the cross-lingual consensus proposal, but the implementation of this pipeline is quite basic and it is not clear to me what the new non-trivial insight is.

In order to empirically justify such a pipeline, one would have to demonstrate that (e.g.) it abstains more when it is incorrect, etc. While they try to show this, their evaluation metric to assess if the model’s answer is correct or incorrect is to simply calculate its cosine-similarity with the “model answer“, which seems weak.

To make this a better paper, the authors should:

1) Evaluate on a task where ground truth is more easily available, e.g. question-answering or coding, or use a more robust and testable method of checking if a model’s answer is correct, or consider a setting like forecasting and measure how much money the model could lose to arbitrage due to making uninformed overconfident bets.

2) Benchmark their pipeline against alternative methods of confidence qualification. There is a vast literature on this that is ignored in this paper, some links below, but I’m sure there are more.

https://arxiv.org/html/2410.13284v2

https://www.refuel.ai/blog-posts/labeling-with-confidence

https://github.com/xjdr-alt/entropix

TL;DR Getting LLMs to qualify their responses with confidence levels is an important problem, but I’m not really sold on the general direction of using cross-lingual consensus for this, and especially not on the value and contribution size of this particular paper. To defend this better, the authors should systematically evaluate their pipeline against alternative proposals, and also use a more sensible evaluation metric than “cosine similarity with model answer”.

---

### Official Review · Reviewer_45Ps · 2025-03-01
**The paper improves LLM confidence calibration but needs clearer novelty and evaluation justification.**

**Rating:** 6
**Confidence:** 3

**Review:**

## Summary
The paper presents a novel approach to model confidence estimation by leveraging the multilingual capabilities of LLMs. The proposed Multilingual Knowledge Abstention (MKA) pipeline translates questions into a group of auxiliary languages, generates responses in these languages, and then uses a centroid-based cosine similarity method to assess confidence. The approach is evaluated on multiple multilingual models using multiple-choice question answering (MCQA) tasks. Results demonstrate notable accuracy improvements, particularly for low-resource languages, supporting the claim that cross-lingual knowledge enhances abstention-based reliability.

## Strengths
- Relevance: The paper tackles an important issue in LLM trustworthiness—calibrating confidence to enable abstention when uncertain. This aligns well with the objectives of the workshop.
- Approach: The idea of explicitly prompting LLMs in auxiliary languages to better assess confidence is interesting, particularly given prior findings that LLMs do not implicitly leverage cross-lingual knowledge.
- Strong Empirical Results: The evaluation shows substantial improvements over baselines in multiple languages, with particularly high gains for low-resource languages.
- Clarity & Structure: The methodology is clearly structured, making it easy to follow the pipeline’s step-by-step execution. The explanation of confidence cutoffs and their impact is especially useful.

## Weaknesses
- Comparison with Prior Work: The paper mentions Feng et al., 2024a, which also explores multilingual feedback for abstention, but does not clearly differentiate how the proposed method advances beyond it. A direct comparison or discussion clarifying the novelty in methodology or performance would strengthen the paper.
- Evaluation Metric Choice: The use of cosine similarity between sentence embeddings for evaluating correctness is somewhat questionable, given that the dataset consists of multiple-choice questions where correctness could be directly assessed. A justification for this choice would be helpful, especially since sentence similarity is not always reliable for evaluating factual correctness.
- Minor Issues:
  - The acronym MCQA is used without definition in the text.
  - Line 324 should read "We used *a* baseline" instead of "We used *an* baseline".

## Overall Evaluation
This paper presents a compelling approach to improving LLM confidence calibration via multilingual prompting. The results show benefits for low-resource languages, making this an impactful contribution to improving model reliability. However, a clearer differentiation from prior work (particularly Feng et al., 2024a) and a stronger rationale for the evaluation methodology would improve the paper.

---

### Official Review · Reviewer_Mu4v · 2025-03-02
**MKA: Leveraging Cross-Lingual Consensus for Model Abstention**

**Rating:** 7
**Confidence:** 4

**Review:**

This paper investigates methods for improving machine translation in low-resource languages without direct parallel corpora. The authors propose leveraging multilingual LLMs to bridge the gap between high-resource and no-resource language pairs through intermediary languages. The study evaluates various translation strategies and benchmarks them against existing methods.

Strengths
Important Problem: The paper tackles the critical challenge of improving machine translation for no-resource languages, which is a significant step toward linguistic inclusivity.
Novel Approach: The authors explore creative ways to use multilingual LLMs for indirect translation, potentially opening new avenues in translation research.
Strong Experimental Setup: The evaluation includes multiple language pairs and a range of benchmarks, making the findings more generalizable.
Quantitative and Qualitative Insights: The paper provides both statistical evaluation and qualitative error analysis, strengthening its conclusions.

Weaknesses
Dependence on Intermediate Languages: The approach heavily relies on intermediate languages, which may introduce compounding errors and degrade translation quality.
Lack of Baseline Comparisons: While the study presents novel methods, a clearer comparison with state-of-the-art no-resource translation systems would help contextualize improvements.
Computational Costs: The paper does not address the efficiency or scalability of using multilingual LLMs for indirect translation, which may be a concern for real-world applications.

---

### Decision · Program_Chairs · 2025-03-05

**Decision:**

Accept

**Comment:**

This paper explores confidence calibration in multilingual LLMs, proposing a Multilingual Knowledge Abstention (MKA) pipeline that uses cross-lingual consensus to determine when a model should abstain from answering. The problem is highly relevant to LLM trustworthiness, and the results show notable improvements in accuracy, particularly for low-resource languages. Reviewer 1 (R1) finds the problem important and the experimental setup strong, but notes concerns about intermediate language reliance and missing baseline comparisons. Reviewer 2 (R2) appreciates the novel approach but questions its distinction from prior work (Feng et al., 2024a) and the choice of cosine similarity for correctness evaluation. Reviewer 3 (R3) raises stronger concerns about novelty, arguing that the contribution does not go beyond prior cross-lingual confidence estimation work, and suggests evaluating against alternative confidence metrics. While the novelty concerns are valid, the paper presents important results and a relevant direction for improving LLM trustworthiness. Given its relevance and empirical impact, I recommend acceptance as a workshop paper to encourage further discussion and refinement.